# Individual Resilience Interventions: A Systematic Review in Adult Population Samples over the Last Decade

**DOI:** 10.3390/ijerph18147564

**Published:** 2021-07-16

**Authors:** Mafalda Ferreira, António Marques, Paulo Veloso Gomes

**Affiliations:** Psychosocial Rehabilitation Laboratory, Center for Rehabilitation Research, School of Health, Polytechnic Institute of Porto, 4200 Porto, Portugal; mafalda.ferreira.mesus@gmail.com (M.F.); pvg@ess.ipp.pt (P.V.G.)

**Keywords:** resilience, interventions, individual, adults, mental health, systematic review

## Abstract

Resilience interventions have been gaining importance among researchers due to their potential to provide well-being and reduce the prevalence of mental disorders that are becoming an increasing concern, especially in Western countries, because of the costs associated. The purpose of this systematic review is to identify the intervention studies carried out in the last decade in adult population samples, evaluate their methodological quality and highlight the trends of these types of interventions. This review was performed using systematic literature searches in the following electronic databases: B-on, PubMed, PsycNet and Science Direct. The application of eligibility criteria resulted in the inclusion of 38 articles, of which 33 were randomized controlled trials and the other five were nonrandomized controlled studies. Although most studies showed statistical significance for their results, these were constrained by the great heterogeneity of the studies, the lack of power of the samples and their fair methodological quality. Therefore, it is important to consolidate the theoretical basis and standardize certain methodologies so that the effects of the interventions can be compared through a meta-analysis.

## 1. Introduction

Resilience emerged as a construct in the physics field to describe the property of certain materials that allows them to absorb the energy of a given impact and then return to a normal state after that pressure is applied [1,2,3,4].

Later, the concept of resilience extended to other areas of knowledge, including psychology. A possible approximation of the construct in the two fields would be to say that the impact or pressure described in physics can be compared to a potentially traumatic situation from which the individual could recover and return to the previous functional state [4]. The definition proposed by the American Psychological Association evidences the connection between the two fields, describing this construct as the ability to “bounce back” in adverse situations [5]. Nevertheless, since psychology is not an exact science, the definition of resilience as a human phenomenon is not so consensual. Some authors define resilience from a more static perspective [6,7] in the sense of recovery to the previous level of functioning, while other authors also adopt, in addition to this aspect, an evolutionary perspective that assumes a level of higher functioning following the adverse event [8,9,10].

Synthesizing some of these perspectives, we can state that resilience can be analyzed on a continuum [11]. A lower level of resilience is a response that assumes a recovery of the functional state after a moderate impact caused by an adverse event. The intermediate level is characterized by a more flexible response, where the impact caused by the adverse event is minimal and short-lived. Both levels correspond to a static type of resilience. Finally, the highest level of resilience consists of a minimal or null impact from which the individual grows and creates a higher level of functioning. This level is also called positive psychological growth [12] and falls under evolutionary resilience. However, can resilience only be defined or developed as a response to an adverse event, as some authors defend [13]? Authors such as Carmello [14], Gonçalves [9] and Pfefferbaum et al. [15] countered this statement by saying that resilience has both reactive and proactive elements. These bold elements can be nurtured and trained and are called the fourth level or strategic resilience. In this case, the individual becomes able to anticipate situations, transform their outcome and become active in the resilience process. From this perspective, together with the increasing prevalence of traumatic and stressful events, resilience interventions become relevant.

According to World Mental Health (WMH) surveys, during a lifetime, more than half of the world’s population (about 70%) experiences at least one potentially traumatic event [16,17,18]. As the experience of a traumatic event is already quite prevalent in today’s society, it is essential to keep in mind that these data assess only life-threatening situations, serious injury or sexual violence [19]. Still, other factors may not be intense enough to be categorized as traumatic, but exposure to them (prolonged or not) can severely affect an individual’s mental health. A significant development in broadening the notions of what potentiates trauma was the change made in the Diagnostic and Statistical Manual of Mental Disorders (DSM-5) criteria that started to consider regular and intense exposures, usually experienced in certain occupations, as a potential inducer of trauma [19].

New concerns are emerging in this regard. The higher prevalence of mental health disorders in the Western culture [17] is inherently allied to the progressive stress loads experienced by individuals in the various spheres of their lives and the difficulties they have in reconciling them [2]. All these aspects lead to an increasing deterioration of mental health that harms the individual and society in general. A growing concern in preventing and treating mental health is due to its high direct costs (associated with treatments) and indirect costs that translate into a lack of productivity, work absences and social exclusion, among others [11,15,20,21]. As a result, some studies have tried to understand if it is possible to develop or strengthen resilience, and, if so, if this would positively impact the prevention and treatment of mental health disorders.

As shown in Figure 1, there has been an upward trend in publications of resilience intervention studies in the last decade, mainly due to studies conducted on specific population groups, such as high-risk occupations or people with severe medical conditions. A good example supporting this data is the Chmitorz et al. [22] study that included all resilience intervention studies in adults from 1974 to 2014, totaling 43 studies. In the present systematic review, the time interval used was significantly shorter (10 years) and totaled 38 studies. Although the inclusion criteria may be slightly different, it is possible to draw some conclusions from these numbers that support this trend.

This study aims to conduct a systematic review of the existing literature on resilience intervention studies conducted on adult population samples to develop or increase individuals’ resilience. This review aims to assess the quality of these studies, draw some conclusions about certain aspects of the design of these interventions and provide information that may be useful for those who may want to develop this field of research.

## 2. Materials and Methods

The present systematic review was conducted in conformity with the Preferred Reporting Items for Systematic Reviews and Meta-analyses (PRISMA) guidelines [23].

### 2.1. Search Strategy

During February of 2021, searches were performed in the following electronic databases: B-on, PubMed, PsycNet and Science Direct. The whole search strategy is available in Appendix B. Advanced search strategies allowed a restriction on the years of publication for the 2010–2020 time interval and language restriction to English, Spanish and Portuguese. In addition to the research carried out in the electronic databases, the list of references of the selected articles was also consulted.

### 2.2. Eligibility Criteria

The eligibility criteria for the review were: (1) randomized or nonrandomized controlled trials assessing the efficacy of resilience interventions; (2) enhancing, promoting or developing individual resilience in adults (<18 years) had to be one of the objectives/aims or hypothesis of the intervention and (3) the use of a valid and appropriate measure of resilience as one of the outcome measures. As there is no standard measure to assess resilience, the authors considered the systematic review of Windle et al. [24] to access scales in terms of validity. In cases in which the scale used was not in this review, the study was included if it proved that the scales used were accurate, described them in detail and provided data on the reliability and validity performed either by the authors or by referencing other studies applied in similar populations.

The exclusion criteria for the present review were: (1) studies that explored the resilience construct in specific contexts or possible associations with other constructs but were not interventions; (2) studies that only described the program, like protocols, but an intervention study was not found or the dates of publication were out of the range; (3) studies that only did a qualitative analysis of the intervention; (4) studies addressed to children, adolescents or elders; (5) studies which exclusively used resilience surrogate outcome measures; (6) studies that did not use a valid resilience scale as an outcome measure; (7) studies that did not target individual resilience and (8) studies where the full text was not available.

### 2.3. Study Selection

An initial screen of the titles and abstracts was performed independently to eliminate papers ineligible to this review. Following this phase, potential eligible articles were retrieved, and the full text was examined, and the inclusion criteria were applied to select the eligible studies. All processes described were conducted using Mendeley Reference Manager Software (Elsevier. London, UK).

### 2.4. Data Collection

A data extraction sheet was developed using Excel, and the data of the selected studies were collected independently. A second author checked the extract data, and refinements to the spreadsheet were discussed. The information extracted from the studies included: author, year of publication, setting, study design, the moment of intervention concerning stress exposure, population, sample size and demographics, study objective/hypothesis, theoretical frame (including resilience definition), description of the intervention and comparator (if applicable), including duration of the program, course of sessions and type of delivery, time points of measurement (including length of follow-up), resilience outcome measures and effectiveness findings. No authors were contacted for further information, since all the data meant for extraction was available. Lack of reporting or unclearly reported data was mentioned in the quality assessment, thus contributing to the poor methodological quality of some studies.

### 2.5. Quality Assessment

According to the Downs and Black Checklist, the methodological quality of all the included studies was assessed individually [25]. In congruence with previous studies [11], the score initially proposed for question 27, “Did the study have sufficient power to detect a clinically important effect where the probability value for a difference being due to chance is less than 5%?” underwent a small change. Instead of the five possible scores presented by the original authors, the results were altered to 0 or 1, based on whether the authors conducted a power analysis to detect a significant clinical effect (of at least 0.80, with alpha at 0.05) or not, with a score of 0 meaning “no” and one meaning “yes”. The ratings of all 28 items were either yes (=1) or no/unable to determine (=0), except for item 5, in which the scores varied as yes (=2), partially (=1) and no (=0). Classification of the final scores fell into four categories: excellent (26–28), good (20–25), fair (15–19) and poor (14 and less).

## 3. Results

### 3.1. Study Selection

The study flow diagram is presented in Figure 2. The electronic search of the databases retrieved 1142 citations that turned into 1050 after duplicates were removed. Through title and abstract screening, 136 studies were identified as potentially eligible, and full-text versions were reviewed. This process resulted in the inclusion of 38 trials. A complete list of the full text papers reviewed and rationale for exclusion is provided in Appendix A. Of the 38 trials included, 33 were randomized controlled trials (RCT), and the other five were nonrandomized controlled studies (NRCS) [26,27,28,29,30,31,32,33,34,35,36,37,38,39,40,41,42,43,44,45,46,47,48,49,50,51,52,53,54,55,56,57,58,59,60,61,62,63].

### 3.2. Study Characteristics

The detailed summary of the studies’ characteristics is described in detail in the data extraction sheet presented in Appendix A.

#### 3.2.1. Setting

In terms of setting, 17 studies were taken in North America (15 of which were specifically in the United States), 11 studies took place in Asia, 8 in Europe, 2 in Oceania and 2 in South America. The high costs related to mental health have led to the need to make decisions regarding the allocation of resources for the implementation of interventions in this area. Since developed countries, particularly Western and some Asian countries, report the highest prevalence and incapacity caused by these diseases [17], this may justify the fact that there are more intervention studies carried out in these countries. Allied to this factor, a possible convenience in implementing these programs in developed countries due to the greater structural and financial predispositions and the available human resources, may also be important in justifying these numbers. The multiculturalism and, therefore, the greater cultural tolerance, can also be an attractive factor for the implementation of programs in these countries, since the cultural barrier will be easily overcome.

However, another fact to denote is that, despite the prevalence of mental health stigma in Asian countries [65], mostly due to cultural reasons, it is also noteworthy that about 29% of the studies were conducted in Asia. This can be a possible indicator that mental health is receiving more attention in Asia, even though there is a lot of regional discrepancy in these efforts.

#### 3.2.2. Population

Since the studies presented a wide variety of populations, they were grouped according to potential risk sources. A suggested grouping for these study populations is shown in Table 1.

Looking at the data, we can conclude that intervention studies on individual resilience in adults, in the last ten years, have been more focused on concerns with stress loads related to occupations and severe medical conditions. Since most of the studies were occupation-related, it should be noted that six of them were workplace resilience interventions. After some consideration, the authors decided to include these studies, because they targeted the individual’s resilience and not their resilience as workers, despite being conducted in a work context. Another conclusion to be drawn is that there were still very few intervention studies applied to more general population samples.

#### 3.2.3. Study Groups

Concerning control groups, twenty-one interventions used waitlist or no intervention groups as comparators, and nine studies compared the main intervention with the usual care condition. As for those that used attention control groups, three of them applied active comparators [46,51,63], while the other five used passive comparison groups [32,33,53,56,57].

Focusing on the main intervention, five studies used multiple groups for this effect: two of these studies compared resilience interventions with alternative methods [57,59], while three studies compared different delivery methods of the same program [31,33,34,36,60,63]. Only one study used a dismantling design to understand each component of the intervention [57].

#### 3.2.4. Objectives

Only nine studies addressed enhancing or developing resilience as a second or third objective and/or hypothesis and used a valid resilience measure as a secondary outcome measure [26,34,35,46,49,51,54,56,63], concluding that the vast majority of studies had resilience as their main focus.

#### 3.2.5. Theoretical Framework and Treatment Approach 

The intervention programs used different approaches: 13 studies used mixed-treatment approaches, while 18 used a single-treatment approach. Seven of the studies used theoretical models to develop their intervention programs. Mindfulness-based interventions, Cognitive Behavioral Therapy (CBT) and Attention and Interpretation Therapy (AIT) were the most common treatment approaches, and the Transactional Model of Stress and Coping [66] was the theoretical model most referenced by the studies.

#### 3.2.6. Resilience Definition

The resilience definitions presented by the studies reviewed are shown in Table 2. An important finding is that about one-third of the studies did not offer any explicit definition of resilience, while two presented more than one. Of the latter studies, one combined a trait–outcome–process approach and the other a trait–process focus. Seventeen studies adopted a trait-oriented approach, and six studies adopted a process-oriented perspective. Seven of the studies considered resilience as a protective factor.

#### 3.2.7. Outcome Measures

Concerning resilience measures, most studies (21/38) used The Connor-Davidson Resilience Scale (CD-RISC) [67], with five of them using the short 10-question version [68]. Both scales understand resilience as a personality trait and assess a compound of resilience factors. The Resilience Scale (RS) [69], evaluating resilience as a personality trait, was the second-most used scale (11/38). Two studies used The Resilience Scale for Adults [70], which is also an assessment tool of protective factors. The other four used The Brief Resilience Scale [71], which assesses resilient outcomes. All these scales and their psychometric properties were described in detail by Windle et al. [24]. In that review, all mentioned scales had a quality rating of 5 or higher, and, although the scoring system ranged from 0 (low) to 18 (high), the maximum score in this review was 7.

Since most studies used a resilience scale that measures resilience as a personality trait and/or resilience factor(s), the findings on the effectiveness of the interventions must be interpreted to a limited extent, given that it can only be translated through a resilient outcome after stress exposure [22].

According to the recommendations presented in previous studies [22], resilience intervention studies should also use measures of stress, as this is a prerequisite of resilience. In addition, measures of mental health are also necessary to obtain additional information about training effects. Considering these recommendations, it was observed that 28 studies assessed measure(s) of mental health, and only 18 studies used (perceived) stress measure(s). About a third of the studies used both as outcome measures. The predominant scale used to measure perceived stress was the Perceived Stress Scale (PSS) [72], and, regarding mental health measures, no study used a measure of overall mental health, with most noting only one or a small number of dysfunctions (22) and only six measuring a larger number of dysfunctions.

#### 3.2.8. Measurement Time Points and Length of Follow-Up

All the studies presented at least two measurement time points: baseline and postintervention. Six studies also measured the outcomes at an intermediate point during the intervention. Most of the studies measured long-term effects, four of them using two or more measurement time points for this purpose. The duration of follow-up ranged from 1 to 12 months.

#### 3.2.9. Moment of Intervention Concerning Stress Exposure

Most studies applied the intervention program during exposure to stress (32/38), with the primary sources of stress in these studies being occupational or disease-related stress. Three interventions applied a preventive perspective in general populations, and three implemented a treatment approach (after stress exposure), mainly in survivor samples.

According to a proposed scheme by Chmitorz et al. [22], in preventive (pre-exposure) studies, only healthy populations should be included, since the objective of these interventions is to protect, not to treat. Among the three studies in this review that were conducted pre-exposure, two used only healthy populations [32,38], while the other did not limit the criteria based on the existence of dysfunction [28].

#### 3.2.10. Intervention Details

Regarding the duration of the intervention programs, nine studies used an eight-week period, while another nine used a 12-week period. The second time frame most applied was five weeks, with the remaining studies varying widely in duration, the shortest duration observed being 4 days and the longest 18 months.

With concerns to the sessions, 22 programs used a weekly frequency to hold the sessions. Other designs presented were retreats, with an additional two studies using a full-day retreat as an additional application to the main intervention type, every other week, a single session, once a day and every three months. Six of the studies did not specify how often the sessions are applied.

About session length, 24 studies provided sessions lasting between one and two hours. Similar to the previous topic, the duration of the sessions varied considerably among the programs, with the shortest duration observed being 10 min and the longest being 3 h.

Concerning the type of delivery, most programs opted for a single method, while six studies opted for a mixed method. Within the studies that applied a single method, 26 used an in-person application. Out of these, 21 applied these face-to-face sessions in groups, and the other five applied them individually. Four of the studies used an online approach, via websites or apps, and two of them used phone calls only. Meanwhile, within the studies that applied mixed methods, two used both in-person sessions and smartphone apps, two used self-administered methods and phone calls and two used both in-person sessions and phone calls.

#### 3.2.11. Effectiveness

In total, 25 of the 38 studies revealed statistically significant data to confirm their hypotheses. Most studies did not report a power analysis. However, since these were mostly feasibility studies and, therefore, used small sample sizes, these analyses may not have been considered. Still, the fact that two-thirds of the studies found statistically significant results supporting their interventions encourages the implementation of large-scale studies, such as RCTs.

#### 3.2.12. Methodological Quality

The detailed process of the quality assessment for each study is presented in Appendix A.

In summary, most of the studies were of fair quality (22). Six studies demonstrated poor quality and ten good quality. None of the studies achieved excellent methodological quality. Overall, the reporting category was the one with the best results. The external validity category was the worst-performing, as it was most often impossible to determine the corresponding data. Regarding the last question related to the power of the studies, this calculation was often not observed, as many studies were intended only for preliminary results for future application on a larger scale.

Only five studies did not perform a comparability analysis across groups at the baseline regarding the demographic variables and the study’s primary outcome measures. Of the 33 who did conduct this analysis, only 14 performed adequate adjustments to confound in the final analyses.

Regarding adverse event reporting, only four studies considered this factor.

## 4. Discussion

### 4.1. Summary of Findings

The field of resilience, particularly its area of intervention, has been gaining importance among researchers and investors due to its potential to provide well-being and the ability to reduce the prevalence of mental dysfunctions. Many studies have proven that the prevention and treatment of mental illness is associated with predictive factors of resilience, such as the purpose and meaning in life, realistic optimism and goal creation, amongst others [73,74]. These factors include a greater variety of dimensions, from individual characteristics to context variables, including family sphere, social context and cultural and normative particularities [75], because resilience is a process shaped by interactions between the individual, those around them and the environment [76]. In this way, interventions that focus on the development of these capabilities will simplify the process of positive adaptation to potentially disruptive events. In other words, resilience can be learned and trained, and, therefore, understanding how it interacts with mental distress is essential to promote population-level mental health [73,76].

As the present systematic review shows, this area of research is a long way from being consolidated. Consequently, the application of these interventions on larger scales is not yet viable. These studies do not follow stricter guidelines in their implementation, evaluation and reporting, making objective and specific analyses challenging to perform. Therefore, the conclusions about the effectiveness and comparisons of methods are of limited confidence.

Given the goal of drawing some conclusions about the latest resilience interventions designed for adult populations, and despite the variability of the various aspects of the studies, it was possible to conclude that there are still significant methodological flaws that can become potential obstacles to the homogenization and consolidation of guidelines for this type of intervention.

In terms of setting, the United States predominance in this area of intervention is clear. Regarding the populations studied, risk groups due to occupation and clinical conditions are also prevalent, while studies conducted in general populations are still very few.

When analyzing the study groups, most studies used waitlist or no intervention conditions for control groups, with only three studies applying active attention control groups. Given that many of the included studies focused on the feasibility and preliminary efficacy of the interventions and, therefore, had small samples, it is natural that the majority of studies opted for less rigorous control groups would be sufficient to guarantee internal validity at the first level. This internal validity, when using no-treatment control arms, implies that it is possible to ensure that participants do not receive any type of treatment. However, without clinical care attention, it is very difficult to ensure the participants’ compliance with treatment, which can affect the results of the study [77]. In addition, if there is any type of care already in place for that population, it is not possible to apply this type of control group. Therefore, many studies have adopted usual care control groups that use treatments that are already in practice as comparators to check whether the new interventions have beneficial effects relative to those already in use, without incurring very high costs in the execution of the studies. Still, this type of control group may also pose threats to the internal validity, since, if the participants are recruited from different populations, this can represent a potential bias, because the treatments may vary due to who provides them and where they are provided [77]. In relation to active control groups, these only become relevant as studies progress and a comparison between two interventions becomes necessary. The reason for this is that it is more difficult to detect the efficacy of one type of treatment in these studies, and they require larger samples due to the inverse relationship between effect sizes and the degree of control [77]. In the case of these types of interventions where clinical care is an essential component and, therefore, does not represent an explanation for the effects, active control groups are often avoided if the intention is not to compare treatments, as they represent a high cost.

Concerning the theoretical framework and treatment approach, the results are more mixed; although almost half of the studies adopted a single-treatment approach, about a third used mixed approaches for the intervention group. Once the efficacy of an intervention has been demonstrated in an RCT, and when their designs comprise several components, a dismantling study should be conducted to identify the components that are effective and show greater efficacy, so that the intervention can be simplified and improved without losing its efficacy [22].

In terms of resilience definition, it is noteworthy that about one-third of the studies did not present any definition, and almost half adopted a view of resilience as a personality trait. Considering that the definitions given should meet those that are the theoretical basis of the scales used [22], it is to be expected that the most used scales were the Connor-Davidson Resilience Scale and the Resilience Scale, both measuring resilience as a personality trait. Still bearing in mind that there should be this bridging in terms of the theoretical frame, it is important to emphasize that nine studies [28,31,32,33,40,41,60,61,63] did not adopt the same approaches both in their definition and scale used.

Regarding the details of the interventions, the methods used also varied significantly. The duration of the interventions ranged from 4 days to 18 months, and the most frequently durations were between 8 and 12 weeks. Session frequencies showed more consistent results, with most studies adopting a weekly frequency. It should be kept in mind that resilience, being a process that can be developed, takes some time to generate results and requires a particular training frequency. Naturally, most studies have adopted a duration between 2 and 3 months, with a weekly frequency. Still, the influence of these factors on the effectiveness of the interventions should be analyzed in more detail in a future study. Regarding the duration of the sessions, a trend was also observed, as 24 studies provided sessions between 1 and 2 h. As for the method of delivery, most interventions opted for a single method that consisted predominantly of face-to-face group sessions.

Most interventions were delivered during stress exposure. This means that the main focus of these interventions was currently placed on preventing mental health problems in populations targeted by long-term exposure. The preventive approach in more generalized populations is still scarce, as is the use of these types of interventions to treat pre-existing mental health disorders.

Regarding the interventions carried out before stress exposure, none of the three used a second postintervention measurement time point after exposure, because the stress event was not considered, and one of them did not measure the long-term efficacy (follow-up). This makes the evaluation of the intervention’s effectiveness impossible, since the resilient outcome was never assessed. Further analyzing the measurement time points, it was observed that 22 of the studies carried out a long-term efficacy evaluation (follow-up), four of which had a duration of only one month.

Overall, most studies provided statistically significant data to confirm their hypotheses. This could mean that there is room to develop these programs on a larger scale and, in parallel, to increase their methodological rigor, especially with regard to their conduct and report power analyses. The quality of most of the studies was fair, with no studies demonstrating excellent quality.

### 4.2. Limitations of the Studies 

As Luthar et al. [78] and, more recently, Chmitorz et al. [22] pointed out, there are still severe gaps in resilience studies, particularly in intervention studies. As it is still an emerging field and a subjective phenomenon, there is a need to consolidate several aspects, such as unifying the concept itself so that future studies can be conducted more systematically. 

As it was possible to observe, many intervention studies still do not present a definition of resilience that supports its theoretical basis. Those that show a definition are quite heterogeneous, leading to a great subjectivity of the interpretations and imposes a major obstacle to comparisons between studies. Another limitation related to the definition and measures of resilience that are used is the fact that most studies still adopt a trait-based perspective. Assuming that resilience is a personality trait gives it a stable character that can hardly be malleable. Such a view does not fit the logic of an intervention that aims to change the resilience of individuals. Thus, it is not possible to conclude that improvements in resilience scores necessarily translate into resilient individuals in the face of an adverse event. For this reason, it is more correct to say that there are indeed many resilience factors (which can also be assessed in studies but as a second objective) but that the aim of intervention studies should be to determine the effectiveness of the intervention and, therefore, to assess the resilience outcome in the face of a stressful event after the intervention. Additionally, in line with this thought, the studies must incorporate mental health measures and relate those with the stress measures.

Another limitation of the studies, in line with what has been observed in previous reviews, is the small sample sizes (most of the times not calculated a priori) limiting the power of the studies that, most of the time, were nor reported by the authors.

An additional limitation found was related to the characteristics of the control groups. Since most studies used a waitlist or no intervention control groups, it was difficult to conclude whether the results found were due to the intervention itself or effects merely related to the attention received. When using control attention groups, some studies also failed to match the treatment doses by comparing an active intervention group with a passive control group.

Concerning the methodological quality of the studies, the authors found three main limitations: the lack of report on the representativeness of the sample, the lack of consideration of the main confounders in the final analysis and the lack of consideration for the possible adverse effects that the intervention could cause.

The lack of follow-up or too-short periods used for these measurements makes it difficult to analyze and compare the effects of long-term interventions.

### 4.3. Strengths and Limitations of This Review

The strengths of this review are the extensive detail of the procedures performed to reach the results that have been reported and the thorough methodological quality assessment of all the studies using a rigorous checklist. Another strength of the present systematic review is the large quantity, quality and usefulness of the information provided. The relevance of the topic and the restriction to studies just from the last decade also allowed a comparison of the developments in this field through other studies and an understanding of the current trends.

This review’s limitations consisted of the narrow scope of the search strategy (only four databases), lack of access to some studies that might be relevant, no consultation of grey literature and a lack of contact with the authors due to time constraints. Another limitation consisted of the large number and, consequently, the significant heterogeneity of the studies that did not allow the quantification of effects and their comparisons.

### 4.4. Recommendations for Future Studies

When it comes to resilience intervention studies, it is essential to consolidate the theoretical foundations before moving on to feasibility studies or even controlled trials. In this sense, Chmitorz et al. [22] presented in their review guidelines to conduct these types of studies.

Regarding treatment approaches, studies should use single approaches or conduct a dismantling design to find out which components most influence the effectiveness of the interventions. Additionally, the doses applied between the intervention and control groups should be comparable concerning treatment. As for the application methods, a balance must be found, bearing in mind that the interventions must be designed to meet the needs of the population for which they are intended. It would also be interesting to focus on more prevention studies in general populations in the near future.

It is also important that the studies measure efficacy in a more longitudinal sense (especially in at-risk populations) and measure the possible adverse effects of interventions.

As far as systematic reviews of resilience intervention studies are concerned, it is vital that they follow a validated protocol/statement and that they delimit the studies to be analyzed well so that a meta-analysis can be performed.

## 5. Conclusions

This systematic review aimed to synthesize the resilience intervention studies conducted in the last decade aimed at adults and focused on individual resilience. 

In this last decade, four systematic reviews (two of them meta-analyses) of these types of studies were published. In 2014, the systematic review and meta-analysis of Leppin et al. [20] and the systematic review of Macedo et al. [79] (which only addressed nonclinical populations) and, in 2018, the systematic review and meta-analysis of Joyce et al. [11] and the systematic review of Chmitorz et al. [22], which provided an interesting framework for future studies and suggestions, were taken into account in the analysis conducted in the present systematic review. Although considering these four systematic reviews mentioned, the current systematic review is relevant in the sense that publications of this type of study have been growing, and, for this reason, 23 of the 38 present studies were not mentioned in any systematic review to our knowledge.

Considering the 38 studies mentioned in this review, it is possible to come to a conclusion on some of the aspects of the latest resilience interventions for adult populations regarding the characteristics of the studies and their potential in promoting well-being. Even so, the significant heterogeneity of the designs, definitions, measures, control groups and other factors made it problematic to conduct a meta-analysis.

## Figures and Tables

**Figure 1 ijerph-18-07564-f001:**
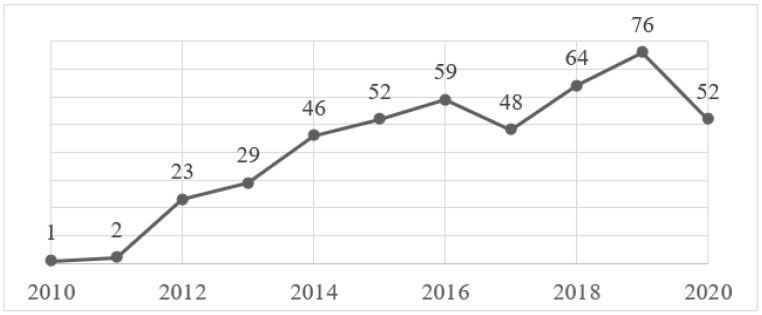
Timeline results of resilience intervention publications by year (2010–2020). Adapted from PubMed.

**Figure 2 ijerph-18-07564-f002:**
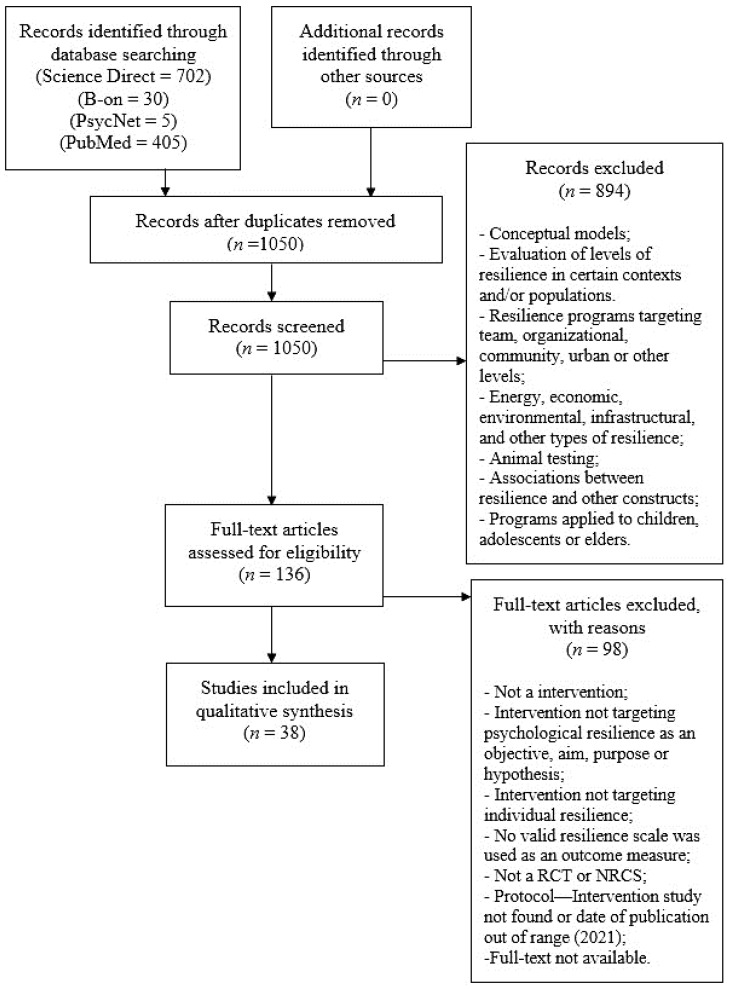
Preferred Reporting Items for Systematic Reviews and Meta-Analyses (PRISMA) flow chart. Adapted from Moher et al. [64].

**Table 1 ijerph-18-07564-t001:** Study population groups.

Group	Studies
General Population	
General population	Champion et al. [38]; Hwang et al. [32]
General population in a specific context	Hendriks et al. [28]
Vulnerable adults	Weiss et al. [51]
New immigrants	Yu et al. [57]
Groups Considered at Risk Due to Occupation	
Employees Emergency workers (general) Law enforcement officers Military recruits Firefighters Female sex workers Human service professionals Health care workers (general) Physicians Nurses College students	Aikens et al. [26]; Kim et al. [33] Wild et al. [53] Christopher et al. [49] Fikretoglu et al. [62] Denkova et al. [59] Wong et al. [54] Pidgeon, Ford & Klaassen [45] Lee et al. [36] Mache et al. [40]; Sood et al. [48]; Sood et al. [50] Hsieh et al. [31]; Lin et al. [37]; Mealer et al. [43] Clarkson at al. [58]; Erogul et al. [61]; Houston et al. [30] Roig et al. [60]
Groups Considered at Risk Due to Clinical Condition	
Workers with chronic health condition	McGonagle, Beatty & Joffe [42]
Individuals with multiple sclerosis	Giovannetti et al. [63]; Senders et al. [46]
Depressed individuals with multiple sclerosis	Kiropoulos et al. [34]
Individuals with Congenital Heart Disease	Kovacs et al. [35]
Breast cancer survivors	Loprinzi et al. [39]; Ye et al. [56]
Breast cancer patients doing chemotherapy	Wu et al. [55]
Cancer patients doing infusion therapy	Mondanaro et al. [44]
Individuals with depression	Songprakun & McCann [47]
Individuals with panic disorder	Wesner et al. [52]
Veterans with PTSD	Burton, Qeadan & Burge [27]
Groups Considered at Risk Due to Caretaking	
Primary caregivers of family members with depression	McCann, Songprakun & Stephenson [41]
Parents at risk	van Grieken et al. [29]

**Table 2 ijerph-18-07564-t002:** Approaches and definitions of the resilience constructs.

Approach	Study	Definition
Trait-Oriented	Burton, Qeadan & Burge [27]	“ability of individuals to adapt positively in the face of trauma” (p. 16)
	Clarkson et al. [58]	“hardiness and ability to cope in adversity (Foureur, Besley, Burton, Yu & Crisp, 2013)” (p. 90)
	Denkova et al. [59]	“ability to effectively adapt to adverse situations (Fletcher and Sarkar, 2013, 2016; Joyce, Shand, Tighe, Laurent, Bryant & Harvey, 2018)”; “malleable characteristic that can be trained and bolstered (Joyce et al., 2018)” (p. 1); “key protective factor” (p. 5)
	Houston et al. [30]	“ability to positively adapt in the face of adversity, trauma or stress (Masten, 2001)” (p. 1)
	Lin et al. [37]	“competency to cope and adapt in the face of adversity, is considered a significant protective factor against the negative effects of job stress (Hart, Brannan, & De Chesnay, 2014)” (p. 118)
	Loprinzi et al. [39]	“ability to thrive despite stress and adversity (Connor & Davidson, 2003)”; “described as invulnerability and hardiness (Kobasa, 1979)”; “the source of resilience is an individual’s innate strength that helps the individual adapt to stressors and pursue life’s meaning and purpose” (p. 365)
	Mache et al. [40]	“individual protective factors such as resilience”; “ability of an individual to withstand adversity and is often seen as a form of self-recovery with positive emotional and cognitive outcomes, which in turn has an important role in realising greater adaptability and life satisfaction (Luthar, Cicchetti & Becker, 2000; Rutter, 1999)” (p. 693)
	McGonagle, Beatty & Joffe [42]	“positive adaptability or ability to thrive in the face of adversity (Campbell-Sills & Stein, 2007; Luthans, 2002)” (p. 387)
	Mealer et al. [43]	“psychological characteristic that has been defined as a trait or capacity depending on the underlying theory adopted”; “one of the most important factors in successful adaptation following exposure to a traumatic event (Charney, 2004)” (p. 98)
	Pidgeon, Ford & Klaassen [45]	“competence to cope and adapt in the face of adversity and to bounce back when stressors become overwhelming is considered a significant protective factor against instances of compassion fatigue, burnout and mental and physical illness (Thomas & Otis, 2010)” (p. 1)
	Songprakun & McCann [47]	“psychosocial capacity of the person to maintain positive adaptive functioning which minimises negative thoughts and promotes recovery of strength and coping ability and to have a positive outlook in the face of difficult circumstances (Reivich, Gillham, Chaplin & Seligman, 2005)”; “protective factor that facilitates successful coping in conditions of adversity (Fergus & Zimmerman, 2005)” (p. 2)
	Sood et al. [48]	“ability of an individual to withstand adversity (Connor & Davidson, 2003)” (p. 858)
	Wesner et al. [52]	“individual’s competence in overcoming stressful life events and adversities (Rutter, 2012)” (p. 428)
	Wild et al. [53]	“capacity to maintain wellbeing in response to adversity or stress (Carleton, Afifi, Turner, Taillieu, Duranceau, et al. 2017)” (p. 2)
	Wong et al. [54]	“the ability to adapt and function competently after adversity” (p. 230)
	Ye et al. [56]	“capacity to bounce back after encountering a traumatic event (Connor and Davidson, 2003; Haglund et al., 2007)” (p. 1487)
	Yu et al. [57]	“effective coping and adaptation when one experiences loss, hardship, or adversity (Tugade & Fredrickson, 2004)” (p. 138)
Process-Oriented	Erogul et al. [61]	“thought to be a state rather than a trait, meaning it is mutable in response to experience or training instead of an innate quality that is fixed and not subject to modification” (p. 353)
	Hsieh et al. [31]	“process of adapting well in the face of adversity, trauma, tragedy, threats, stress, serious health problems, or workplace conflict—it means “bouncing back” from difficult experiences (APA, 2014)” (p. 2)
	Hwang et al. [32]	“the process of adapting well in the face of adversity, trauma, tragedy, threats or even significant sources of threat (APA)” (p. 2); “factor that potentially buffers against the negative impact of work stress (Howard, 2008)” (p. 5)
	Kim et al. [33]	“refers to the process that allows individuals to adapt positively despite stress or trauma (Luthar, Cicchetti & Becker, 2000)” (p. 8)
	McCann, Songprakun & Stephenson [41]	“process of coping with adversity, change, or opportunity in a manner that results in the identification, fortification, and enrichment of resilient qualities or protective factors (Richardson 2002, p. 308)” (pp. 62,63)
	Roig et al. [60]	“may be understood as the personal assets (internal factors, e.g., optimism) and environmental resources (external factors, e.g., social support) that contribute to positive psychological adaption, despite exposure to adversity (Helmereich, Kunzler, Chmitorz, König, Binder & Wessa, 2017)” (p. 2)
Trait-Process	Giovannetti et al. [63]	“an internal resource for alleviating the adverse effects of stress and sustaining good mental health through adversity (Leppin, Bora, Tilburt, Gionfriddo, Zeballos-Palacios, Dulohery, et al., 2014). It entails the process of negotiating, managing and adapting to significant stressors or trauma through drawing on internal (…), and external (…) resources (Windle, Bennet & Noyes, 2011)” (p. 2)
Trait-Process-Outcome	Hendriks et al. [28]	“capacity to deal effectively with stress and adversity, to adapt successfully to setbacks (Luthar, Cicchetti, & Becker, 2000; Zautra, Hall, & Murray, 2008), and to bounce back after negative emotional experiences (Tugade & Fredrickson, 2004). Resilience refers to positive outcomes in spite of threats to adaptation or development (Masten, 2001) and factors and mechanisms that play a role in dealing functionally with, and contribute to successful adaptation to problems (Friborg, Hjemdal, Martinussen, & Rosenvinge, 2009). Resilience is an active process” (p. 2); “set of personal characteristics” (p. 3)

## Data Availability

All the data analyzed in this review are included in the present article.

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
