# Peer review of "Individual Resilience Interventions: A Systematic Review in Adult Population Samples over the Last Decade"

_ijerph, 2021, doi:10.3390/ijerph18147564_

Round 1
Reviewer 1 Report
I consider that there are phrases in the text that are conjecture or are based on subjective elements:
"When analyzing the study groups, the supremacy of waitlist and no intervention conditions for control groups is also notorious, as well as the use of active attention control groups by only three studies. This indicates that the conclusions of these studies are limited since the groups being compared cannot be matched on type and dose of treatment".
"Concerning the theoretical framework and treatment approach, the results are more mixed: although almost half of the studies adopted a single-treatment approach, about a third used mixed approaches for the intervention group. The use of mixed-treatment approaches can also be limiting if a dismantling technique is not used since the more techniques used, the harder it is to conclude which intervention components contribute to the results".
More recent citations should be added based on aspects of resilience that are capable of making the reader understand the relationship of this concept with the central theme of the research.
Author Response
Dear Reviewer,
Thank you for your valuable contribution.
We send a new version of the article, in word format, with the changelog activated, with the added or altered parts in a different color for easy reference and reading.
To view point-by-point response to comments, please see the attachment.
All suggestions were considered. We hope to have met expectations.

Reviewer 2 Report
First of all, I would like to appreciate that I was able to review the meaningful research, entitled: "Individual Resilience Interventions: a systematic review in adult population samples over the last decade." Despite the limited number of studies regarding resilience interventions, the current research found significant findings based on prior studies. Especially, the systematic review that were conducted by the authors was scientific, which could be considered a major strength of this article. Assuming its potential contribution to the mental health areas, I, as a reviewer, suggest the following comments to maximize the strength of article.
- pp 5, line 155: The author mentioned that "The prior statement about growing concerns for mental health in Western cultures is supported by these results." However, reader could see that 11 out 0f 38 (more than 28% - can't be ignored) are Asian studies. Can author suggest criteria to justify this issue?
- pp 13, line 430: Regarding the following statements, "Considering the 38 studies mentioned in this review, it is possible to conclude on some of the main trends of resilience interventions for adult populations regarding the characteristics of the studies and their potential in promoting well-being," - I assume that heterogeneity of the sample could be found and it would not be easy to find "main trends" when the samples' characteristic is different. Can authors suggest a basis of generalizability based on the sample groups that have different background?
- Related to the issue #2, can the author add list of independent and control variables in the appendix section that were used in 38 studies?
Author Response

(The authors gave the same response as above.)
